# Mechanical and Crack-Sensing Capabilities of Mode-I Joints with Carbon-Nanotube-Reinforced Adhesive Films under Hydrothermal Aging Conditions

**DOI:** 10.3390/nano10112290

**Published:** 2020-11-19

**Authors:** Xoan F. Sánchez-Romate, Jesús Martin, María Sánchez, Alejandro Ureña

**Affiliations:** Materials Science and Engineering Area, Escuela Superior de Ciencias Experimentales y Tecnología, Universidad Rey Juan Carlos, Calle Tulipán s/n, 28933 Móstoles, Spain; j.martincord@alumnos.urjc.es (J.M.); maria.sanchez@urjc.es (M.S.); alejandro.urena@urjc.es (A.U.)

**Keywords:** nanostructures, smart materials, fracture toughness, joints, joining

## Abstract

The fracture behavior and crack sensing of mode-I joints with carbon nanotube (CNT)-reinforced adhesive films were explored in this paper under hydrothermal aging conditions. The measured fracture energy of CNT-reinforced joints in grit blasting conditions is higher for non-aged samples than for neat adhesive joints (around 20%) due to the nanofiller toughening and crack bridging effects. However, in the case of brushed surface-treated adherents, a drastic decrease is observed with the addition of CNTs (around 70%) due to the enhanced tribological properties of the nanofillers. Hydrothermal aging has a greater effect in the CNT-reinforced samples, showing a more prevalent plasticization effect, which is confirmed by the R-curves of the specimens. The effects of surface treatment on the crack propagation properties was observed by electrical resistance monitoring, where brushed samples showed a more unstable electrical response, explained by more unstable crack propagation and reflected by sharp increases of the electrical resistance. Aged specimens showed a very uniform increase of electrical resistance due to slower crack propagation, as induced by the plasticization effect of water. Therefore, the proposed adhesive shows a high applicability for crack detection and propagation without decreasing the mechanical properties.

## 1. Introduction

Carbon nanoparticles, and more specifically carbon nanotubes (CNTs), are now gaining attention due to the exceptional properties that they exhibit in terms of electrical and thermal conductivity, as well as mechanical strength [1,2,3]. In this regard, they are widely used as reinforcing elements for polymers. In fact, they allow the creation of electrical networks inside these insulating media, enhancing the electrical conductivity by several orders of magnitude [4,5,6,7,8].

This enhancement of electrical properties can give new functionalities to the enhanced materials. For example, the creation of percolating networks allows their use in structural health monitoring (SHM) applications by means of electrical conductivity measurements [9,10]. Their use is based on the fact that the electrical resistance between adjacent nanoparticles, which is called the tunneling resistance, increases in a linear exponential way with the interparticle distance [11,12], making these CNT nanocomposites very sensitive to the applied strain [13,14,15,16]. In addition, these materials have also demonstrated very good capabilities for determining damage, as the presence of cracks induces the breakage of electrical pathways inside the network, causing a related increase of the electrical resistivity [17,18].

Furthermore, the addition of CNTs into a polymeric matrix induces an enhancement of the mechanical properties at very low contents due to the toughening mechanisms of the CNTs themselves [5,19]. Here, the dispersion of nanofillers plays a crucial role, as the presence of larger agglomerates can lead to premature failure of the material. In this context, the effects of different dispersion techniques such as ultrasonication, three roll milling, and toroidal and mechanical stirring on the dispersion state of the CNTs have been widely explored [20,21,22]. It has been proven that dispersion not only plays a critical role in terms of the mechanical properties, but also in terms of the electrical or thermal properties, as the presence of aggregates can reduce the efficiency of electrical and thermal CNT networks [23,24].

Moreover, CNTs have also been demonstrated to be an effective reinforcements for adhesive joints. In fact, their addition usually leads to an increase of the lap shear strength (LSS), and specifically the mode-I fracture energy, due to the previously commented toughening effect of CNTs, as well as their good crack-bridging capabilities [25,26,27,28,29]. In addition, they can act as effective corrosion barriers [30]. Furthermore, their applicability for SHM purposes has been demonstrated, having excellent sensitivity to both strain and crack propagation mechanisms [31,32,33]

In this regard, this work aims to study the crack propagation mechanisms in bonded joints using a novel CNT-reinforced adhesive film under aging conditions. Simultaneously, the electrical monitoring capabilities will also be explored to better understand the mechanical behavior of the joints. The SHM capabilities of the proposed adhesive films have been proved under quasi-static and fatigue conditions in previous studies [34,35,36].

The main novelty of the proposed work is the analysis of how the addition of CNTs affects the crack propagation in mode-I joints under aging conditions, a fact that remains to be investigated. The effects of water uptake have been previously explored in SLS joints [37], where an enhancement of the mechanical properties was observed in CNT-reinforced adhesives when compared to neat ones. However, crack propagation mechanisms under aging conditions have not been explored, as the failure in SLS joints takes place very rapidly.

For this purpose, several mode-I joints with aluminum alloy substrates in pristine state and after several days of water immersion were manufactured. Here, apart from the aging conditions, the effects of the surface treatment of the adherent were also explored. In every case, electrical resistance was simultaneously measured during the mechanical tests to observe the correlation between electrical properties and mechanical performance.

## 2. Experimental

### 2.1. Materials and Manufacturing

The adhesive film used in this work was FM 300 K film supplied by (Cytec, Woodland Park, NJ, USA), with a nominal thickness of 0.2 mm. It had a knit tricot carrier allowing for good thickness control. This material is adequate for metal–metal joints.

NC7000 multiwall carbon nanotubes (MWCTNs) were supplied by (Nanocyl, Sambreville, Belgium). They had lengths of up to 1.5 μm and diameters measuring 9.5 nm. A sodium dodecyl sulphate (SDS) surfactant was used for stabilization of the dispersion.

The manufacturing of CNT-reinforced adhesive films was carried in three steps: (i) 0.1 wt.% CNT dispersion in an aqueous solution with 0.25 wt.% SDS by ultrasonication for 20 min, as previously optimized [37,38] in order to ensure the good electrical conductivity of the adhesive film; (ii) spraying of the CNT dispersion over the adhesive using an airbrush at a pressure of 1 bar and positioning over the aluminum substrate; (iii) curing in a hot press following the curing cycle outlined in Table 1.

Mode-I energy fracture tests were carried out by following the ISO 25217:2009 standard for aged and non-aged samples. The dimensions of the substrates were 150 × 25 × 4 mm^2^, with a pre-crack length of 65 mm, which was ensured by placing Teflon prior to curing. The bond line thickness was 0.3 mm. Three specimens were tested for each condition.

Aging was carried out by immersion of the whole joint in a distilled water bath at 60 °C for 15 days to promote a rapid water diffusion effect similar to a previous work [39]. Aluminum substrates were subjected to two different surface treatments: brushing with SiC P120 sandpaper and grit blasting at 6 bar of pressure with corundum particles of 1 mm diameter. The aim of using a brushing treatment, despite it not being widely used in practical applications, was to demonstrate the sensitivity of the proposed technique to surface preparation. Table 2 summarizes the nomenclature for the samples tested. Three specimens were tested for each condition and the error bars were calculated using the standard deviation procedure.

### 2.2. Electromechanical Tests

Electrical resistance was recorded with an Agilent 34410 *A* module (Agilent Technologies, Santa Clara, CA, USA) at an acquisition frequency of 10 Hz. For this purpose, two electrodes made of copper wire and sealed with silver ink were attached to each substrate, as shown in the schematics of Figure 1a.

The crack length was recorded by a video camera. The edges of the joint were painted in white for better observation. This allowed the main crack propagation mechanisms to be correlated with the measured electrical resistance.

### 2.3. Joint Characterization

The surface topology of the brushed and grit-blasted substrates was analyzed by means of optical profilometry using a ZETA Z-20 instrument (Zeta Instruments, San José, CA, USA). For this purpose, the average roughness, *R_a_*, was determined and optical images were included.

Microstructural characterization of the fracture surfaces of the joints was carried out by scanning electron microscopy (SEM) analysis using a Hitachi 3400N module (Hitachi, Tokyo, Japan). The samples were coated by sputtering with a thin layer of gold to allow proper observation.

## 3. Results and Discussion

In this section, the mechanical performance of mode-I joints was characterized in pristine and aged specimens. The fracture energy values and R-curves are shown. Furthermore, the electrical response as a function of crack length was studied in order to corroborate the SHM capabilities of the proposed materials.

### 3.1. G_IC_ Values at Initial Conditions

The fracture energy in mode-I, *G_IC_*, of the adhesive joints was calculated using the enhanced compliance method (ECM) [40], as it has higher accuracy than other typical solutions, such as the simple beam theory (SBT) or the corrected beam theory (CBT) [41]:(1)$$G_{IC}\left(\frac{J}{m^{2}}\right)=\frac{n\times P\times \delta }{2\times B\times a}\times \frac{F}{N}\;\;$$
where *P* is the applied load, *B* the width of the adherent, *δ* is the piano hinge displacement, *a* is the crack length starting from the load application point, and *F* and N are two correction factors associated with the load and displacement, which are estimated as follows:(2)$$F=1-\frac{3}{10}\times {\left(\frac{\delta }{a}\right)}^{2}-\frac{3}{2}\times \left(\frac{\delta \times l_{1}}{a^{2}}\right)$$
(3)$$N=1-{\left(\frac{l_{2}}{a}\right)}^{3}-\frac{9}{8}\times \left(1-{\left(\frac{l_{2}}{a}\right)}^{2}\right)\times \frac{\delta \times l_{1}}{a^{2}}-\frac{9}{35}\times {\left(\frac{\delta }{a}\right)}^{2}$$

The parameters *l*_1_ and *l*_2_ are as indicated in the schematics of Figure 1b.

Moreover, the *n* factor of Equation (1) is calculated as the slope of the logarithm of compliance, *C =*
*δ/P*, divided by *N* versus log*(a)*, as follows:(4)$$n=slope\left(\frac{\log\left(C/N\right)}{\log\left(a\right)}\right)$$

According to these expressions, Figure 2 summarizes the average values of *G_IC_* during the first 25 mm of crack propagation for the tested conditions at the initial stage (non-aged state). Here, it can be observed that there are two opposite trends when comparing neat joints versus CNT-reinforced joints, depending on the surface treatment. On the one hand, for grit-blasted specimens, there is an increase of the fracture energy value when adding CNTs. This effect has been previously observed in Carbon fiber to carbon fiber (CFRP-CFRP) joints [34] and is explained by the toughening and crack bridging effect of the CNTs [25,42]. However, when comparing brushed specimens, the addition of CNTs causes a drastic decrease of the fracture energy value. Here, the enhanced tribological properties of CNTs [43] play a crucial role in the interfacial adhesion, causing a very drastic decrease in this case. This can be explained by the roughness values of the adherents. In the case of brushed specimens, the average roughness is much lower (0.457 μm) than for grit-blasted specimens (1.53 μm), as well as being less homogeneous, as can be observed in the optical profilometry images in Figure 3. Therefore, the higher the average roughness value and uniformity, the more prevalent the creation of a more active surface allowing better adhesion of the epoxy adhesive [44].

In this regard, Figure 4 shows images of the fracture surfaces for the different tested conditions. Here, it can be observed that cohesive failure is the main failure mode in every case except for the CNT-reinforced brushed (CNT-B) joints (left image of Figure 4b), which show a very prevalent adhesive failure (as can be observed in the detailed left image of Figure 4b highlighted in red). Therefore, this is indicative of the poor interfacial adhesion, which explains the low values of the fracture energy. The analysis of the fracture surfaces by SEM also confirmed the prevalence of interfacial failure in the case of CNT-brushed specimens (Figure 5a) in comparison to grit-blasted ones (Figure 5b). However, the neat adhesive samples did not show any significant differences when comparing brushing (Figure 5c) with grit blasting (Figure 5d) surface treatments, explaining the similar values for the fracture energy.

### 3.2. Influence of Aging Mechanisms

Figure 6 shows the average *G_IC_* values during the first 25 mm of crack propagation for the different specimens under aging conditions. Here, it can be observed that the fracture energy values are significantly lower than in the case of non-aged samples, as also summarized in Table 3. This can be explained according to two factors: on the one hand, the water uptake promotes the creation of hydrogen bonds inside the polymeric network, which induces a highly prevalent plasticization effect [45]; on the other hand, a possible decrease of the interfacial adhesion occurs, which the aging process promotes [46].

Water uptake measurements were obtained from previous studies [37,39] for CNT and neat adhesives, and are represented in the graph in Figure 7. Here, it can be observed that water uptake values in equilibrium conditions for CNT-reinforced adhesives and the diffusion coefficient (slope of the initial linear region) are slightly higher than for neat adhesives (1.43 ×10^−5^ and 9.53 × 10^−6^ mm^2^/h, respectively). This is explained by the amphiphilic effect of SDS, whose hydrophilic radical tends to potentiate the water absorption. This effect is more prevalent than the barrier properties of the nanofillers.

When analyzing the influence of CNT addition on fracture energy values, it can be observed that the CNT-reinforced samples present a much more prevalent decrease of *G_IC_* than the neat adhesive joints in the case of grit-blasted specimens. This is explained by the higher plasticization effect induced by the presence of SDS, as previously explained. However, for brushed specimens, the decrease of fracture energy is less significant when adding CNTs. The reason lies in the fact that the interfacial adhesion is very poor in the case of CNT-reinforced non-aged samples, so they do not show a significant decrease when subjected to hydrothermal aging, as the crack propagation mechanisms are ruled by adhesion forces and not by the cohesive behavior of the adhesive itself, meaning the mechanical performance is less affected by the plasticization effect of the water. On the other hand, the neat adhesive samples in non-aged conditions show better interfacial adhesion with cohesive failure, as observed before, so they are more affected by the aging conditions as they are more dominated by the cohesive forces of the adhesive.

This plasticization effect of water uptake can also be corroborated by the R-curves of the specimens, which are shown in the graph in Figure 8. Here, it can be observed that the fracture energy as a function of crack length is almost constant in the case of non-aged specimens (solid symbols), which is indicative of brittle behavior, whereas the aged samples show an increase in the R-curve slope, which is indicative of a more ductile behavior [47]. This trend is not observed for the CNT-reinforced samples with brushing treatment due to the poor interfacial adhesion that dominates the crack propagation behavior.

Furthermore, SEM images of fracture surfaces in Figure 9 highlight the differences between the crack propagation mechanisms in non-aged and aged samples. Here, the samples at the initial stage show generally smoother fracture surfaces (Figure 9a) when compared to the aged specimens, where rougher fracture surfaces are observed (Figure 9b), which indicates more ductile fracture behavior, and thus a plasticization effect.

### 3.3. Crack Propagation Monitoring

Table 4 summarizes the initial values of the electrical resistance, *R*_0_, for the different tested conditions. Here, it can be observed that the surface treatment has a very prevalent effect on the electrical properties of the adhesive joint, as expected. More specifically, the GB joints show much lower *R*_0_ values than brushed ones. This is explained by the better interfacial adhesion, which promotes the creation of more effective electrical pathways [35]. On the other hand, the effect of hydrothermal aging is reflected by an increase of the electrical resistance due to a decrease of the interfacial adhesion, thus leading to weaker contact between the conductive adhesive and the aluminum substrate.

Figure 10 shows some examples of the electromechanical curves of CNT-reinforced mode-I specimens. It can be observed that the electrical resistance increases with the crack length (represented as dots), as there is breakage of the electrical pathways. As noted in previous studies [34,48], three different regions can be distinguished: (i) the first stage, where there is no crack propagation; (ii) an initial sudden increase of the crack length after reaching the peak load; (iii) crack growth until final failure of the specimen.

In region (i), there is no increase of the electrical resistance as there is no breakage of electrical pathways. Region (ii) is mainly characterized by a sudden increase of the electrical resistance, which is associated with the first stages of crack propagation, where it increases very rapidly. Finally, the electrical resistance increases in region (iii) in different ways depending on the crack propagation mechanisms.

In this regard, it can be noticed that the electromechanical behavior is significantly affected by the surface treatment. Here, it can be observed that grit-blasted specimens show a very stable increase of the electrical resistance in region (iii), as stated in Figure 10a. This is explained by more uniform crack propagation, which is also reflected in the mechanical curve, which exhibits a soft decrease of the applied load with piano hinge displacement.

However, brushed specimens show a much more unstable electrical response. In region (iii), the electrical resistance is characterized by sharp increases of the electrical resistance, which correspond to sudden drops in the applied load, as shown in Figure 10b. This can be explained by unstable crack propagation. In fact, each sudden drop is correlated with rapid crack propagation inside the joint. This means that there is prevalent breakage of the electrical pathways, thus explaining the sudden increase of the electrical resistance. This is usually correlated with a poor interface and with the significant presence of defects, inducing typical stick–slip behavior [32,48], as can be observed in the schematics in Figure 11. Furthermore, the crack propagation takes places in a more rapid way in the case of brushed specimens, which is also reflected by a more prevalent increase of the electrical resistance. Therefore, the electrical resistance, in combination with the mechanical performance, can give more detailed information about the crack propagation mechanisms and the quality of the interface.

On the other hand, when comparing non-aged (Figure 10a) with aged specimens (Figure 10c), some interesting facts can be noticed. In both cases, there is an initial crack propagation region with a sudden increase of the electrical resistance due to a prevalence of electrical breakage mechanisms. Then, region (iii) is characterized by a stable increase of the electrical resistance. However, when analyzing in detail the electromechanical responses, some slight differences can be pointed out. On the one hand, the crack propagation takes place in a slower way in the first stages of region (ii) (marked as region iii.1 in Figure 10d) in the case of aged specimens, where some sharp increases of the electrical resistance can be observed in the non-aged samples (red circles in Figure 10d). Here, the increase of the electrical resistance is softer than in the case of non-aged specimens, as the crack propagation is slower. This was previously explained due to the plasticization effect of the hydrogen bonds because of the water uptake and was also noted in the R-curves in Figure 8. This promotion of weak hydrogen bonds leads to a reduction of the stiffness of the epoxy matrix, promoting softer crack propagation. This softer propagation leads to a softer breakage of electrical pathways, which is correlated with slower crack propagation, and thus is reflected by a smooth evolution of the electrical resistance during the crack propagation phase. Furthermore, the electromechanical behavior in the very last stages of mechanical tests (region iii.2 of Figure 10d) is quite similar in both cases, although a sudden increase of the electrical resistance is observed at 17.5 mm in the case of non-aged samples, probably due to a misconnection of electrical pathways in the last stages of the test.

Therefore, the analysis of the electromechanical behavior reveals that the monitoring of the proposed CNT-reinforced adhesive films can give detailed information about the crack propagation mechanisms and allow a better understanding of the effects of both surface treatment and aging conditions. Here, it is worth pointing out that the intrinsic properties of the CNTs themselves also play an important role. Specifically, these are correlated with the overall electrical properties of the final nanocomposite, such as the aspect ratio and functionalization, as they play a prevalent role in the creation of electrical percolating networks.

## 4. Conclusions

The mechanical performance and electromechanical responses of mode-I joints made of neat and CNT-reinforced adhesive films under aging conditions have been widely explored.

The fracture energy of CNT-reinforced samples in grit-blasted conditions is significantly higher than neat ones due to the toughening and crack-bridging effects of the CNTs. However, in brushing conditions, the mechanical performance is drastically affected by the addition of CNTs, with a critical decrease of the fracture energy. This is explained by the enhanced tribological properties of CNTs, promoting a lack of interfacial adhesion between the adhesive and substrate, which is reflected by a main adhesive failure.

The hydrothermal aging behavior of the joints was also investigated. Here, the CNT addition induces a more prevalent decrease of the fracture energy in the case of grit-blasted specimens, as the amphiphilic behavior of the surfactant causing the dispersion leads to slightly greater water absorption than in the neat adhesive. Therefore, the plasticization effect of hydrothermal aging is more prevalent in the CNT-reinforced samples, which is reflected by a reduction of the fracture energy and a stable increase of the slopes of the R-curves. In the case of brushed samples, CNT-reinforced samples do not show a prevalent plasticization effect, as the failure takes place in the interfacial region between the adhesive and substrate.

The electromechanical behavior of mode-I joints shows an increase of electrical resistance with crack length due to the breakage of electrical pathways. Here, grit-blasted samples show a more uniform increase of the electrical resistance due to a more stable crack propagation process, as well as a more uniform surface treatment. However, brushed specimens show typical stick–slip behavior, showing sharp increases of the electrical resistance followed by arrest phases. The effect of hydrothermal aging is reflected by a softer increase of the electrical resistance due to slower crack propagation promoted by the plasticization effect of the water.

Therefore, the electrical monitoring sheds light on the main crack propagation mechanisms, depending on the surface treatment and aging conditions, proving the good potential and applicability of the proposed adhesive films for structural health monitoring purposes.

## Figures and Tables

**Figure 1 nanomaterials-10-02290-f001:**
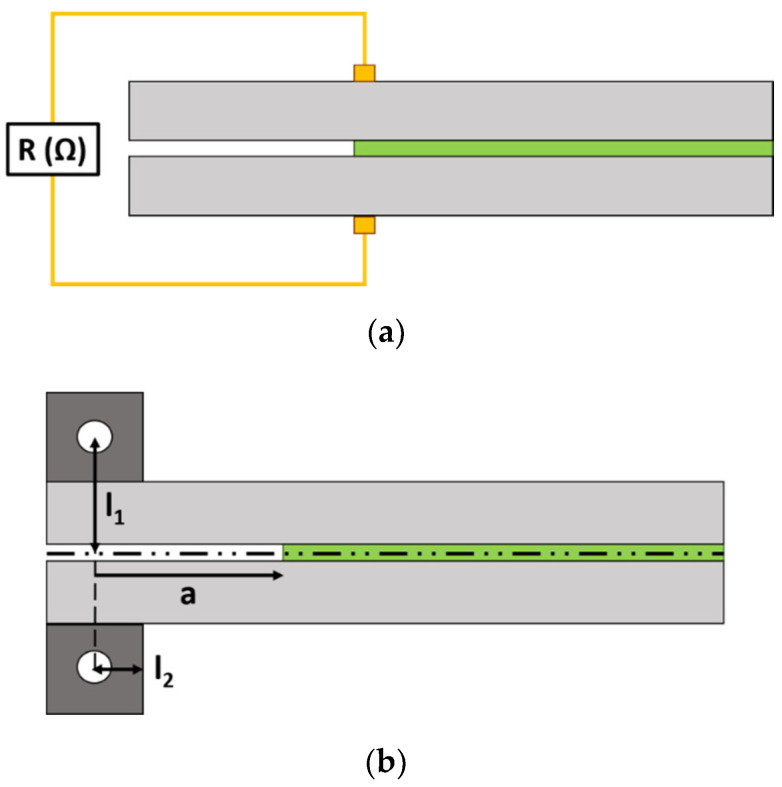
Schematics of the (**a**) electrode and (**b**) piano hinge disposition.

**Figure 2 nanomaterials-10-02290-f002:**
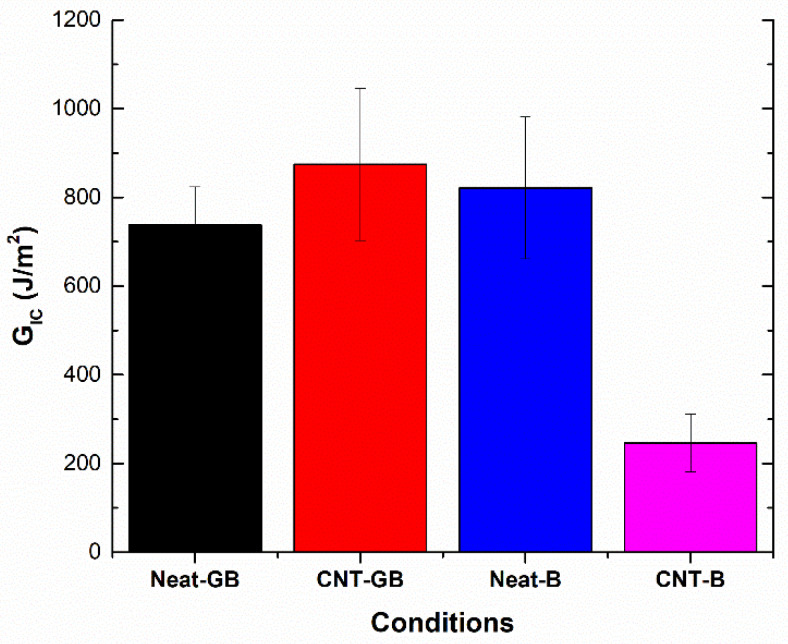
Fracture energy in Mode-I, *G_IC_*_,_ values of non-aged specimens.

**Figure 3 nanomaterials-10-02290-f003:**
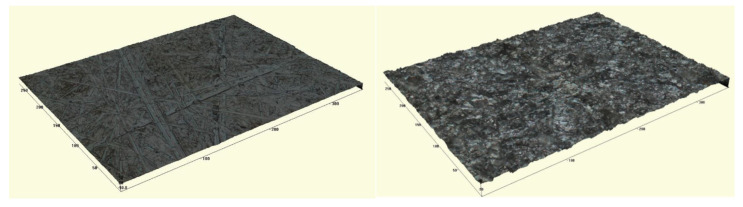
Profilometry images of (**left**) brushed and (**right**) grit-blasted surface treatments on the adherents.

**Figure 4 nanomaterials-10-02290-f004:**
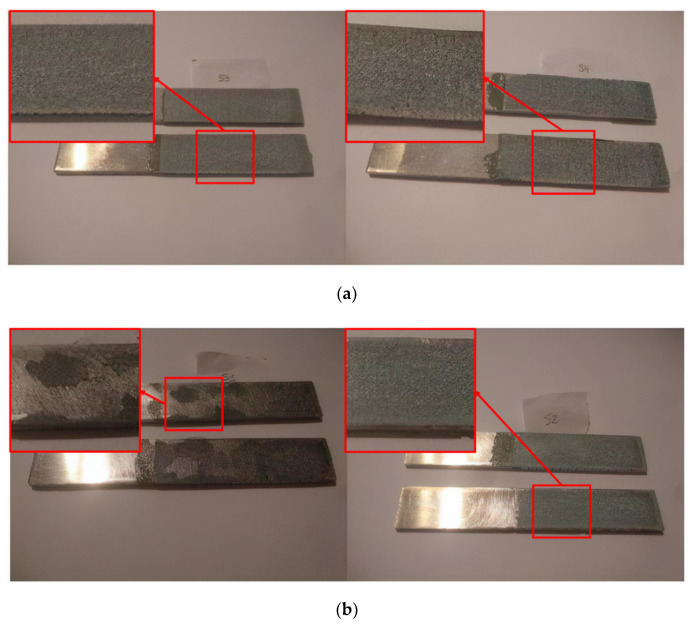
Fracture surface images of: (**a**) non-aged CNT-reinforced grit-blasted (CNT-GB) (**left**) and non-aged neat grit-blasted, GB (**right**); (**b**) non-aged CNT-reinforced brushed (CNT-B) (**left**) and non-aged neat brushed, B (**right**). (The highlighted areas show detailed views of the main failure mode.)

**Figure 5 nanomaterials-10-02290-f005:**
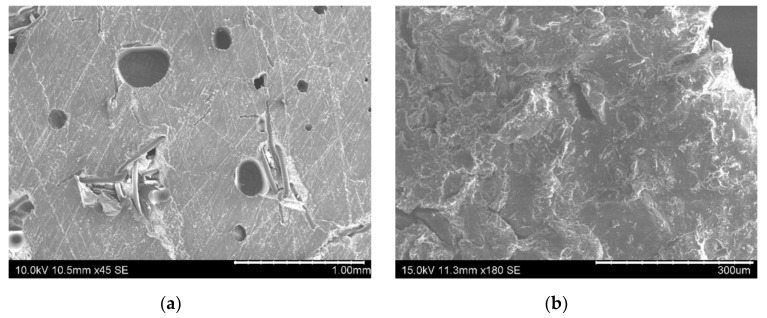
SEM images of fracture surfaces of (**a**) CNT-B, (**b**) CNT-GB, (**c**) neat B, and (**d**) non-aged neat GB samples.

**Figure 6 nanomaterials-10-02290-f006:**
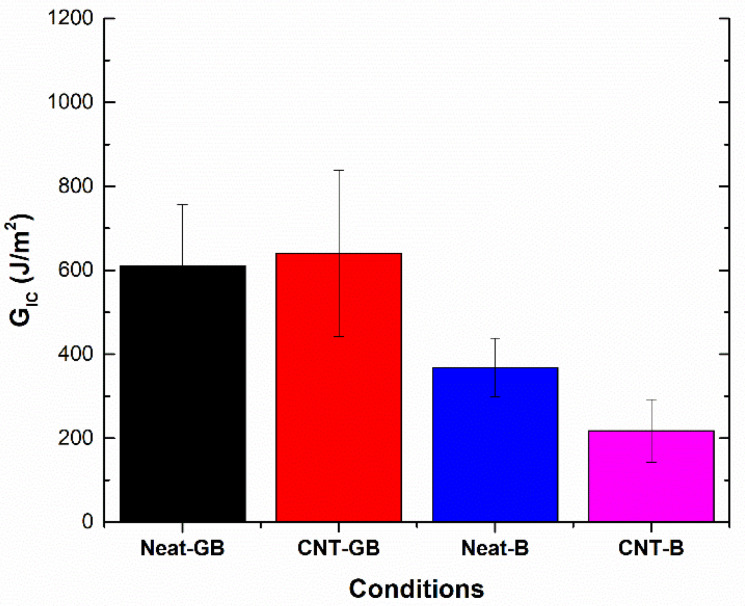
Fracture energy in Mode-I, *G_IC_* values of aged specimens.

**Figure 7 nanomaterials-10-02290-f007:**
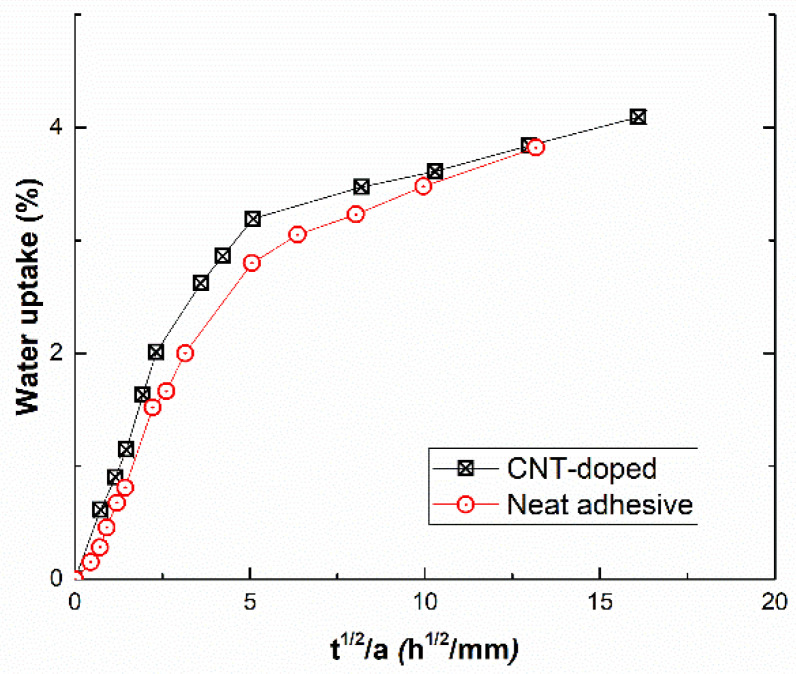
Water uptake of CNT-reinforced and neat adhesives, where *a* denotes the thickness of the specimen and *t* denotes the aging time (data obtained from [39]).

**Figure 8 nanomaterials-10-02290-f008:**
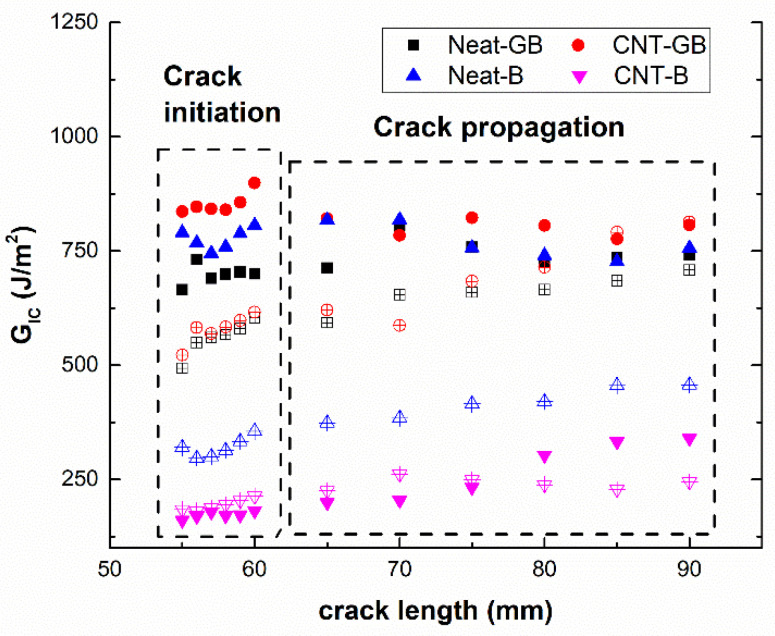
R-curves for the different tested conditions (solid and hollow symbols denote the non-aged and aged samples, respectively).

**Figure 9 nanomaterials-10-02290-f009:**
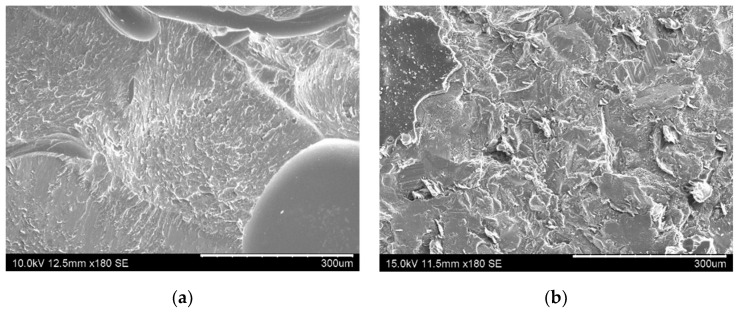
SEM images of the fracture surfaces of (**a**) non-aged and (**b**) aged CNT-reinforced samples in GB conditions.

**Figure 10 nanomaterials-10-02290-f010:**
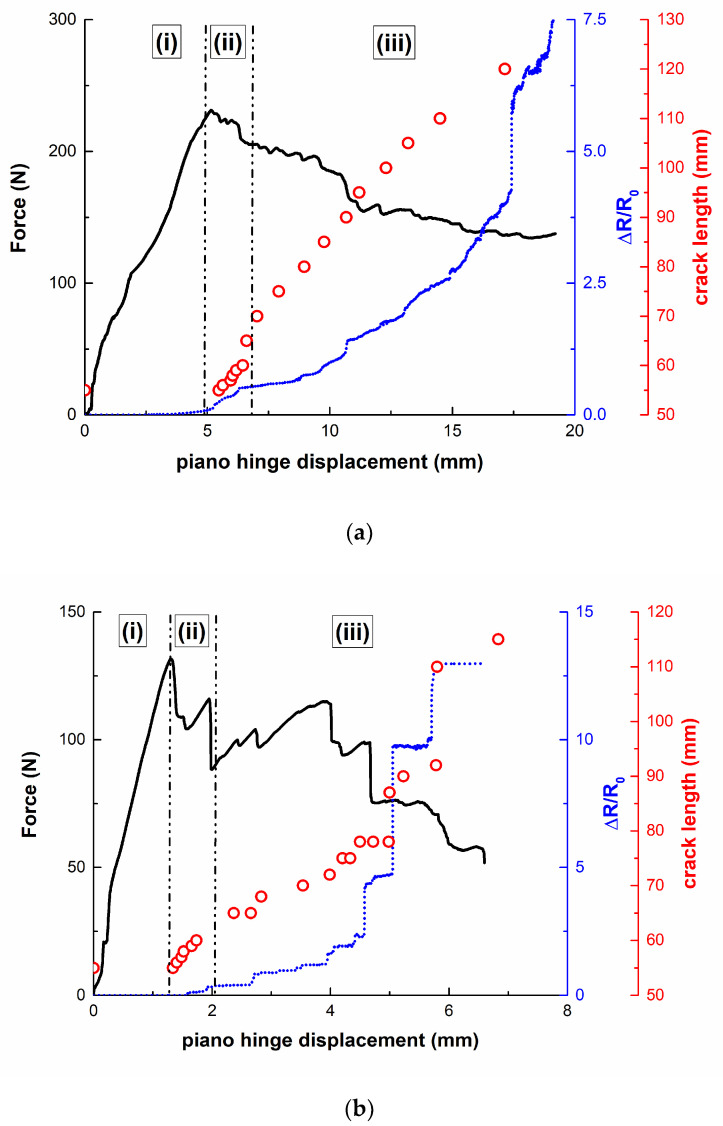
Electrical resistance variation as a function of the applied load and crack length of CNT-reinforced samples, showing differences between (**a**) grit-blasted and (**b**) brushed samples and between (**a**) non-aged and (**c**) aged samples. (**d**) Detailed electromechanical responses of non-aged and aged samples, where the areas highlighted in red represent the points experiencing sudden crack propagation.

**Figure 11 nanomaterials-10-02290-f011:**
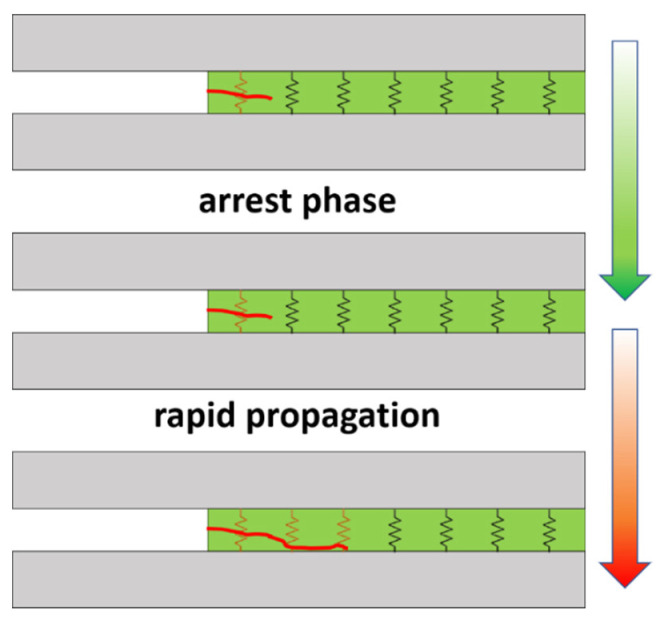
Schematics of crack propagation monitoring and the crack’s effect on the breakage of electrical pathways with typical stick–slip behavior.

**Table 1 nanomaterials-10-02290-t001:** Curing cycle parameters used for bulk adhesive and joints.

Parameter	First Stage	Second Stage
Pressure	Ramp from 0 to 0.6 MPa for 15 min	0.6 MPa for 90 min
Temperature	Ramp from 25 to 175 °C for 45 min	175 °C for 60 min

**Table 2 nanomaterials-10-02290-t002:** Nomenclature for the different tested joints.

Material	Condition	Nomenclature
Neat adhesive	Brushing at initial stage	Neat B non-aged
	Grit blasting at initial stage	Neat GB non-aged
	Brushing after aging	Neat B aged
	Grit blasting after aging	Neat GB aged
0.1 wt.% CNT-reinforced adhesive	Brushing at initial stage	CNT-B non-aged
	Grit blasting at initial stage	CNT-GB non-aged
	Brushing after aging	CNT-B aged
	Grit blasting after aging	CNT-GB aged

**Table 3 nanomaterials-10-02290-t003:** Summary of energy fracture values for the different tested conditions.

Condition	*G_IC_* (J/m^2^)	
	Non-Aged	Aged
Neat GB (grit-blasted)	737 ± 86	610 ± 147
CNT-GB	875 ± 172	640 ± 198
Neat B (brushed)	821 ± 160	369 ± 70
CNT-B	246 ± 65	218 ± 75

**Table 4 nanomaterials-10-02290-t004:** Values of the initial electrical resistance for the different tested conditions.

Surface Treatment	*R*_0_ (Ω)	
	Non-Aged	Aged
0	214 ± 30	433 ± 46
B (Brushed)	329 ± 70	4190 ± 1477

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
