# Peer review of "Mechanical and Crack-Sensing Capabilities of Mode-I Joints with Carbon-Nanotube-Reinforced Adhesive Films under Hydrothermal Aging Conditions"

_nanomaterials, 2020, doi:10.3390/nano10112290_

Round 1
Reviewer 1 Report
Please see the attached file.

Reviewer 2 Report
This work reports Mechanical and Crack Sensing Capabilities of Mode-I joints with Carbon Nanotube-Doped Adhesive Films under Hydrothermal Ageing. The methodology and experimental details were well-written and interesting to the readers. There are some minor points authors should apply for enhanced the quality of their paper :
1- It is better to avoid the CNT-doped term since it stands for the chemical alteration of the CNT structure by introducing foreign atoms
2- How did the authors come with optimal concentration for CNT? Did they try lower and higher comparison for the thickness?
3- Please show Figures 3a and 3b together no separately.
4- Figure 6 and 1 good be combined for better comparison.
5- Can the authors provide which properties of CNTs is more important for their studied properties for example aspect ratio, or degree of oxidation, etc.
Reviewer 3 Report
This study is relevant. It shows damage monitoring in the Mode-I joints using an electrically conductive adhesive and the effect of surface treatment in the crack propagation properties considering also aging effects.
The paper is well organized and the characterization is accurate and well done.
I recommend this work for publication after that following points have been addressed:
1) In the introduction, the effect of the use carbon nanotubes in the adhesives for structural applications, should be introduced in light of works in the literature as
https://doi.org/10.1016/j.matdes.2012.04.052
https://doi.org/10.3390/ma10101131
https://doi.org/10.1016/j.compositesb.2014.09.022
https://doi.org/10.1016/j.compositesb.2017.07.021
2) In all SEM images the size bar should be highlighted.
3) In section 3.2, the calculation of the diffusion coefficient has been only mentioned. The diffusion coefficient values for both systems should be valued and inserted in the manuscript
4) As the authors state, the electrical resistance of the joint for the filled grit-blasted system increases, due to water aging, due to the presence of an additional resistance created in the adhesive / metal interface. For this reason, the variation in electrical resistance during the mechanical test is attributable only to the crack propagation speed, faster in the case of the dry system, slower in the case of the wet system. Plasticization phenomena make the adhesive matrix less rigid, influencing the crack propagation speed.
The comment relating figure 10 should be improved and better explained, including the relationship between plasticization phenomena and DR/R0
